# NOT-MIWAE: DEEP GENERATIVE MODELLING WITH MISSING NOT AT RANDOM DATA

**Niels Bruun Ipsen**[*]
nbip@dtu.dk

**Pierre-Alexandre Mattei**[†‡]
pierre-alexandre.mattei@inria.fr

**Jes Frellsen**[*‡]
jefr@dtu.dk

## ABSTRACT

When a missing process depends on the missing values themselves, it needs to be explicitly modelled and taken into account while doing likelihood-based inference. We present an approach for building and fitting deep latent variable models (DLVMs) in cases where the missing process is dependent on the missing data. Specifically, a deep neural network enables us to flexibly model the conditional distribution of the missingness pattern given the data. This allows for incorporating prior information about the type of missingness (e.g. self-censoring) into the model. Our inference technique, based on importance-weighted variational inference, involves maximising a lower bound of the joint likelihood. Stochastic gradients of the bound are obtained by using the reparameterisation trick both in latent space and data space. We show on various kinds of data sets and missingness patterns that explicitly modelling the missing process can be invaluable.

## 1 INTRODUCTION

Missing data often constitute systemic issues in real-world data analysis, and can be an integral part of some fields, e.g. recommender systems. This requires the analyst to take action by either using methods and models that are applicable to incomplete data or by performing imputations of the missing data before applying models requiring complete data. The expected model performance (often measured in terms of imputation error or innocuity of missingness on the inference results) depends on the assumptions made about the missing mechanism and how well those assumptions match the true missing mechanism. In a seminal paper, Rubin (1976) introduced a formal probabilistic framework to assess missing mechanism assumptions and their consequences. The most commonly used assumption, either implicitly or explicitly, is that a part of the data is *missing at random*

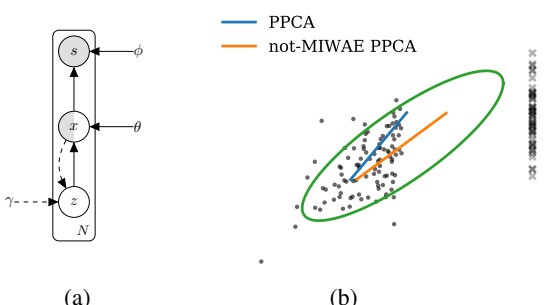

Figure 1: (a) Graphical model of the not-MIWAE. (b) Gaussian data with MNAR values. Dots are fully observed, partially observed data are displayed as black crosses. A contour of the true distribution is shown together with directions found by PPCA and not-MIWAE with a PPCA decoder.

*(MAR)*. Essentially, the MAR assumption means that the missing pattern does not depend on the missing values. This makes it possible to ignore the missing data mechanism in likelihood-based inference by marginalizing over the missing data. The often implicit assumption made in non-probabilistic models and ad-hoc methods is that the data are *missing completely at random (MCAR)*. MCAR is a stronger assumption than MAR, and informally it means that both observed and missing data do not depend on the missing pattern. More details on these assumptions can be found in the monograph of Little & Rubin (2002); of particular interest are also the recent revisits of Seaman et al. (2013) and Doretti et al. (2018). In this paper, our goal is to *posit statistical models that leverage deep learning in order to break away from these assumptions.* Specifically, we propose a general

[*]Department of Applied Mathematics and Computer Science, Technical University of Denmark, Denmark

[†]Université Côte d'Azur, Inria (Maasai team), Laboratoire J.A. Dieudonné, UMR CNRS 7351, France

[‡]Equal contribution

recipe for dealing with cases where there is prior information about the distribution of the missing pattern given the data (e.g. self-censoring).

The MAR and MCAR assumptions are violated when the missing data mechanism is dependent on the missing data themselves. This setting is called *missing not at random (MNAR)*. Here the missing mechanism cannot be ignored, doing so will lead to biased parameter estimates. This setting generally requires a joint model for data and missing mechanism.

Deep latent variable models (DLVMs, Kingma & Welling, 2013; Rezende et al., 2014) have recently been used for inference and imputation in missing data problems (Nazabal et al., 2020; Ma et al., 2018; 2019; Ivanov et al., 2019; Mattei & Frellsen, 2019). This led to impressive empirical results in the MAR and MCAR case, in particular for high-dimensional data.

## 1.1 CONTRIBUTIONS

We introduce the *not-missing-at-random importance-weighted autoencoder (not-MIWAE)* which allows for the application of DLVMs to missing data problems where the missing mechanism is MNAR. This is inspired by the missing data importance-weighted autoencoder (MIWAE, Mattei & Frellsen, 2019), a framework to train DLVMs in MAR scenarios, based itself on the importance-weighted autoencoder (IWAE) of Burda et al. (2016). The general graphical model for the not-MIWAE is shown in figure 1a. The first part of the model is simply a latent variable model: there is a stochastic mapping parameterized by $\theta$ from a latent variable $z \sim p(z)$ to the data $x \sim p_\theta(x|z)$, and the data may be partially observed. The second part of the model, which we call the missing model, is a stochastic mapping from the data to the missing mask $s \sim p_\phi(s|x)$. Explicit specification of the missing model $p_\phi(s|x)$ makes it possible to address MNAR issues.

The model can be trained efficiently by maximising a lower bound of the joint likelihood (of the observed features and missing pattern) obtained via importance weighted variational inference (Burda et al., 2016). A key difference with the MIWAE is that we use the reparameterization trick in the data space, as well as in the code space, in order to get stochastic gradients of the lower bound.

Missing processes affect data analysis in a wide range of domains and often the MAR assumption does not hold. We apply our method to censoring in datasets from the UCI database, clipping in images and the issue of selection bias in recommender systems.

## 2 BACKGROUND

Assume that the complete data are stored within a data matrix $X = (x_1, \ldots, x_n)^\intercal \in \mathcal{X}^n$ that contain $n$ i.i.d. copies of the random variable $x \in \mathcal{X}$, where $\mathcal{X} = \mathcal{X}_1 \times \cdots \times \mathcal{X}_p$ is a $p$-dimensional feature space. For simplicity, $x_{ij}$ refers to the $j$'th feature of $x_i$, and $x_i$ refers to the $i$'th sample in the data matrix. Throughout the text, we will make statements about the random variable $x$, and only consider samples $x_i$ when necessary. In a missing data context, each sample can be split into an observed part and a missing part, $x_i = (x_i^o, x_i^m)$. The pattern of missingness is individual to each copy of $x$ and described by a corresponding mask random variable $s \in \{0, 1\}^p$. This leads to a mask matrix $S = (s_1, \ldots, s_n)^\intercal \in \{0, 1\}^{n \times p}$ verifying $s_{ij} = 1$ if $x_{ij}$ is observed and $s_{ij} = 0$ if $x_{ij}$ is missing.

We wish to construct a parametric model $p_{\theta,\phi}(x, s)$ for the joint distribution of a single data point $x$ and its mask $s$, which can be factored as

$$p_{\theta,\phi}(x, s) = p_\theta(x)p_\phi(s|x). \tag{1}$$

Here $p_\phi(s|x) = p_\phi(s|x^o, x^m)$ is the conditional distribution of the mask, which may depend on both the observed and missing data, through its own parameters $\phi$. The three assumptions from the framework of Little & Rubin (2002) (see also Ghahramani & Jordan, 1995) pertain to the specific form of this conditional distribution:

- MCAR: $p_\phi(s|x) = p_\phi(s)$,
- MAR: $p_\phi(s|x) = p_\phi(s|x^o)$,
- MNAR: $p_\phi(s|x)$ may depend on both $x^o$ and $x^m$.

To maximize the likelihood of the parameters $(\theta, \phi)$, based only on observed quantities, the missing data is integrated out from the joint distribution

$$p_{\theta,\phi}(\boldsymbol{x}^{\mathrm{o}}, \boldsymbol{s}) = \int p_\theta(\boldsymbol{x}^{\mathrm{o}}, \boldsymbol{x}^{\mathrm{m}}) p_\phi(\boldsymbol{s}|\boldsymbol{x}^{\mathrm{o}}, \boldsymbol{x}^{\mathrm{m}}) \, \mathrm{d}\boldsymbol{x}^{\mathrm{m}}. \tag{2}$$

In both the MCAR and MAR cases, inference for $\theta$ using the full likelihood becomes proportional to $p_{\theta,\phi}(\boldsymbol{x}^{\mathrm{o}}, \boldsymbol{s}) \propto p_\theta(\boldsymbol{x}^{\mathrm{o}})$, and the missing mechanism can be ignored while focusing only on $p_\theta(\boldsymbol{x}^{\mathrm{o}})$. In the MNAR case, the missing mechanism can depend on both observed and missing data, offering no factorization of the likelihood in equation (2). The parameters of the data generating process and the parameters of the missing data mechanism are tied together by the missing data.

## 2.1 PPCA EXAMPLE

A linear DLVM with isotropic noise variance can be used to recover a model similar to probabilistic principal component analysis (PPCA, Roweis, 1998; Tipping & Bishop, 1999). In figure 1b, a dataset affected by an MNAR missing process is shown together with two fitted PPCA models, regular PPCA and the not-MIWAE formulated as a PPCA-like model. Data is generated from a multivariate normal distribution and an MNAR missing process is imposed by setting the horizontal coordinate to missing when it is larger than its mean, i.e. it becomes missing because of the value it would have had, had it been observed. Regular PPCA for missing data assumes that the missing mechanism is MAR so that the missing process is ignorable. This introduces a bias, both in the estimated mean and in the estimated principal signal direction of the data. The not-MIWAE PPCA assumes the missing mechanism is MNAR so the data generating process and missing data mechanism are modelled jointly as described in equation (2).

## 2.2 PREVIOUS WORK

In (Rubin, 1976) the appropriateness of ignoring the missing process when doing likelihood based or Bayesian inference was introduced and formalized. The introduction of the EM algorithm (Dempster et al., 1977) made it feasible to obtain maximum likelihood estimates in many missing data settings, see e.g. Ghahramani & Jordan (1994; 1995); Little & Rubin (2002). Sampling methods such as Markov chain Monte Carlo have made it possible to sample a target posterior in Bayesian models, including the missing data, so that parameter marginal distributions and missing data marginal distributions are available directly (Gelman et al., 2013). This is also the starting point of the multiple imputations framework of Rubin (1977; 1996). Here the samples of the missing data are used to provide several realisations of complete datasets where complete-data methods can be applied to get combined mean and variability estimates.

The framework of Little & Rubin (2002) is instructive in how to handle MNAR problems and a recent review of MNAR methods can be found in (Tang & Ju, 2018). Low rank models were used for estimation and imputation in MNAR settings by Sportisse et al. (2020a). Two approaches were taken to fitting models, 1) maximising the joint distribution of data and missing mask using an EM algorithm, and 2) implicitly modelling the joint distribution by concatenating the data matrix and the missing mask and working with this new matrix. This implies a latent representation both giving rise to the data and the mask. An overview of estimation methods for PCA and PPCA with missing data was given by Ilin & Raiko (2010), while PPCA in the presence of an MNAR missing mechanism has been addressed by Sportisse et al. (2020b). There has been some focus on MNAR issues in the form of selection bias within the recommender system community (Marlin et al., 2007; Marlin & Zemel, 2009; Steck, 2013; Hernández-Lobato et al., 2014; Schnabel et al., 2016; Wang et al., 2019) where methods applied range from joint modelling of data and missing model using multinomial mixtures and matrix factorization to debiasing existing methods using propensity based techniques from causality.

Deep latent variable models are intuitively appealing in a missing context: the generative part of the model can be used to sample the missing part of an observation. This was already utilized by Rezende et al. (2014) to do imputation and denoising by sampling from a Markov chain whose stationary distribution is approximately the conditional distribution of the missing data given the observed. This procedure has been enhanced by Mattei & Frellsen (2018a) using Metropolis-within-Gibbs. In both cases the experiments were assuming MAR and a fitted model, based on complete data, was already available.

Approaches to fitting DLVMs in the presence of missing have recently been suggested, such as the HI-VAE by Nazabal et al. (2020) using an extension of the variational autoencoder (VAE) lower bound, the p-VAE by Ma et al. (2018; 2019) using the VAE lower bound and a permutation invariant encoder, the MIWAE by Mattei & Frellsen (2019), extending the IWAE lower bound (Burda et al., 2016), and GAIN (Yoon et al., 2018) using GANs for missing data imputation. All approaches are assuming that the missing process is MAR or MCAR. In (Gong et al., 2020), the data and missing mask are modelled together, as both being generated by a mapping from the same latent space, thereby tying the data model and missing process together. This gives more flexibility in terms of missing process assumptions, akin to the matrix factorization approach by Sportisse et al. (2020a).

In concurrent work, Collier et al. (2020) have developed a deep generative model of the observed data conditioned on the mask random variable, and Lim et al. (2021) apply a model similar to the not-MIWAE to electronic health records data. In forthcoming work, Ghalebikesabi et al. (2021) propose a deep generative model for non-ignorable missingness building on ideas from VAEs and pattern-set mixture models.

## 3  INFERENCE IN DLVMS AFFECTED BY MNAR

In an MNAR setting, the parameters for the data generating process and the missing data mechanism need to be optimized jointly using all observed quantities. The relevant quantity to maximize is therefore the log-(joint) likelihood

$$\ell(\theta, \phi) = \sum_{i=1}^{n} \log p_{\theta,\phi}(\boldsymbol{x}_i^{\mathrm{o}}, \boldsymbol{s}_i), \tag{3}$$

where we can rewrite the general contribution of data points $\log p_{\theta,\phi}(\boldsymbol{x}^{\mathrm{o}}, \boldsymbol{s})$ as

$$\log \int p_{\phi}(\boldsymbol{s}|\boldsymbol{x}^{\mathrm{o}}, \boldsymbol{x}^{\mathrm{m}}) p_{\theta}(\boldsymbol{x}^{\mathrm{o}}|\boldsymbol{z}) p_{\theta}(\boldsymbol{x}^{\mathrm{m}}|\boldsymbol{z}) p(\boldsymbol{z}) \, \mathrm{d}\boldsymbol{z} \, \mathrm{d}\boldsymbol{x}^{\mathrm{m}}, \tag{4}$$

using the assumption that the observation model is fully factorized $p_{\theta}(\boldsymbol{x}|\boldsymbol{z}) = \prod_j p_{\theta}(x_j|\boldsymbol{z})$, which implies $p_{\theta}(\boldsymbol{x}|\boldsymbol{z}) = p(\boldsymbol{x}^{\mathrm{o}}|\boldsymbol{z}) p_{\theta}(\boldsymbol{x}^{\mathrm{m}}|\boldsymbol{z})$. The integrals over missing and latent variables make direct maximum likelihood intractable. However, the approach of Burda et al. (2016), using an inference network and importance sampling to derive a more tractable lower bound of $\ell(\theta, \phi)$, can be used here as well. The key idea is to posit a conditional distribution $q_{\gamma}(\mathbf{z}|\boldsymbol{x}^{\mathrm{o}})$ called the *variational distribution* that will play the role of a learnable proposal in an importance sampling scheme.

As in VAEs (Kingma & Welling, 2013; Rezende et al., 2014) and IWAEs (Burda et al., 2016), the distribution $q_{\gamma}(\mathbf{z}|\boldsymbol{x}^{\mathrm{o}})$ comes from a simple family (e.g. the Gaussian or Student's $t$ family) and its parameters are given by the output of a neural network (called inference network or encoder) that takes $\boldsymbol{x}^{\mathrm{o}}$ as input. The issue is that a neural net cannot readily deal with variable length inputs (which is the case of $\boldsymbol{x}^{\mathrm{o}}$). This was tackled by several works: Nazabal et al. (2020) and Mattei & Frellsen (2019) advocated simply zero-imputing $\boldsymbol{x}^{\mathrm{o}}$ to get inputs with constant length, and Ma et al. (2018; 2019) used a permutation-invariant network able to deal with inputs with variable length.

Introducing the variational distribution, the contribution of a single observation is equal to

$$\log p_{\theta,\phi}(\boldsymbol{x}^{\mathrm{o}}, \boldsymbol{s}) = \log \int \frac{p_{\phi}(\boldsymbol{s}|\boldsymbol{x}^{\mathrm{o}}, \boldsymbol{x}^{\mathrm{m}}) p_{\theta}(\boldsymbol{x}^{\mathrm{o}}|\boldsymbol{z}) p(\boldsymbol{z})}{q_{\gamma}(\boldsymbol{z}|\boldsymbol{x}^{\mathrm{o}})} q_{\gamma}(\boldsymbol{z}|\boldsymbol{x}^{\mathrm{o}}) p_{\theta}(\boldsymbol{x}^{\mathrm{m}}|\boldsymbol{z}) \, \mathrm{d}\boldsymbol{x}^{\mathrm{m}} \, \mathrm{d}\boldsymbol{z} \tag{5}$$

$$= \log \mathbb{E}_{\boldsymbol{z} \sim q_{\gamma}(\boldsymbol{z}|\boldsymbol{x}^{\mathrm{o}}), \boldsymbol{x}^{\mathrm{m}} \sim p_{\theta}(\boldsymbol{x}^{\mathrm{m}}|\boldsymbol{z})} \left[ \frac{p_{\phi}(\boldsymbol{s}|\boldsymbol{x}^{\mathrm{o}}, \boldsymbol{x}^{\mathrm{m}}) p_{\theta}(\boldsymbol{x}^{\mathrm{o}}|\boldsymbol{z}) p(\boldsymbol{z})}{q_{\gamma}(\boldsymbol{z}|\boldsymbol{x}^{\mathrm{o}})} \right]. \tag{6}$$

The main idea of importance weighed variational inference and of the IWAE is to replace the expectation inside the logarithm by a Monte Carlo estimate of it (Burda et al., 2016). This leads to the objective function

$$\mathcal{L}_K(\theta, \phi, \gamma) = \sum_{i=1}^{n} \mathbb{E} \left[ \log \frac{1}{K} \sum_{k=1}^{K} w_{ki} \right], \tag{7}$$

where, for all $k \leq K, i \leq n$,

$$w_{ki} = \frac{p_{\phi}(\boldsymbol{s}_i|\boldsymbol{x}_i^{\mathrm{o}}, \boldsymbol{x}_{ki}^{\mathrm{m}}) p_{\theta}(\boldsymbol{x}_i^{\mathrm{o}}|\boldsymbol{z}_{ki}) p(\boldsymbol{z}_{ki})}{q_{\gamma}(\boldsymbol{z}_{ki}|\boldsymbol{x}_i^{\mathrm{o}})}, \tag{8}$$

and $(\boldsymbol{z}_{1i}, \boldsymbol{x}_{1i}^{\mathrm{m}}), \ldots, (\boldsymbol{z}_{Ki}, \boldsymbol{x}_{Ki}^{\mathrm{m}})$ are $K$ i.i.d. samples from $q_\gamma(\boldsymbol{z}|\boldsymbol{x}_i^{\mathrm{o}})p_\theta(\boldsymbol{x}^{\mathrm{m}}|\boldsymbol{z})$, over which the expectation in equation (7) is taken. The unbiasedness of the Monte Carlo estimates ensures (via Jensen's inequality) that the objective is indeed a lower-bound of the likelihood. Actually, under the moment conditions of (Domke & Sheldon, 2018, Theorem 3), which we detail in Appendix D, it is possible to show that the sequence $(\mathcal{L}_K(\theta, \phi, \gamma))_{K \geq 1}$ converges monotonically (Burda et al., 2016, Theorem 1) to the likelihood:

$$\mathcal{L}_1(\theta, \phi, \gamma) \leq \ldots \leq \mathcal{L}_K(\theta, \phi, \gamma) \xrightarrow[K \to \infty]{} \ell(\theta, \phi). \tag{9}$$

**Properties of the not-MIWAE objective**    The bound $\mathcal{L}_K(\theta, \phi, \gamma)$ has essentially the same properties as the (M)IWAE bounds, see Mattei & Frellsen, 2019, Section 2.4 for more details. The key difference is that we are integrating over *both the latent space and part of the data space*. This means that, to obtain unbiased estimates of gradients of the bound, we will need to backpropagate through samples from $q_\gamma(\boldsymbol{z}|\boldsymbol{x}_i^{\mathrm{o}})p_\theta(\boldsymbol{x}^{\mathrm{m}}|\boldsymbol{z})$. A simple way to do this is to use the reparameterization trick *both for $q_\gamma(\boldsymbol{z}|\boldsymbol{x}_i^{\mathrm{o}})$ and $p_\theta(\boldsymbol{x}^{\mathrm{m}}|\boldsymbol{z})$*. This is the approach that we chose in our experiments. The main limitation is that $p_\theta(\boldsymbol{x}|\boldsymbol{z})$ has to belong to a reparameterizable family, like Gaussians or Student's $t$ distributions (see Figurnov et al., 2018 for a list of available distributions). If the distribution is not readily reparametrisable (e.g. if the data are discrete), several other options are available, see e.g. the review of Mohamed et al. (2020), and, in the discrete case, the continuous relaxations of Jang et al. (2017) and Maddison et al. (2017).

**Imputation**    When the model has been trained, it can be used to impute missing values. If our performance metric is a loss function $L(\boldsymbol{x}^{\mathrm{m}}, \hat{\boldsymbol{x}}^{\mathrm{m}})$, optimal imputations $\hat{\boldsymbol{x}}^{\mathrm{m}}$ minimise $\mathbb{E}_{\boldsymbol{x}^{\mathrm{m}}}[L(\boldsymbol{x}^{\mathrm{m}}, \hat{\boldsymbol{x}}^{\mathrm{m}})|\boldsymbol{x}^{\mathrm{o}}, \boldsymbol{s}]$. When $L$ is the squared error, the optimal imputation is the conditional mean that can be estimated via self-normalised importance sampling (Mattei & Frellsen, 2019), see appendix B for more details.

### 3.1 Using prior information via the missing data model

The missing data mechanism can both be known/decided upon in advance (so that the full relationship $p_\phi(\boldsymbol{s}|\boldsymbol{x})$ is fixed and no parameters need to be learned) or the type of missing mechanism can be known (but the parameters need to be learnt) or it can be unknown both in terms of parameters and model. The more we know about the nature of the missing mechanism, the more information we can put into designing the missing model. This in turn helps inform the data model how its parameters should be modified so as to accommodate the missing model. This is in line with the findings of Molenberghs et al. (2008), who showed that, for MNAR modelling to work, one has to leverage prior knowledge about the missing process. A crucial issue is under what model assumptions the full data distribution can be recovered from incomplete sample. Indeed, some general missing models may lead to inconsistent statistical estimation (see e.g. Mohan & Pearl, 2021; Nabi et al., 2020).

The missing model is essentially solving a classification problem; based on the observed data and the output from the data model filling in the missing data, it needs to improve its "accuracy" in predicting the mask. A Bernoulli distribution is used for the probability of the mask given both observed and missing data

$$p_\phi(\boldsymbol{s}|\boldsymbol{x}^{\mathrm{o}}, \boldsymbol{x}^{\mathrm{m}}) = p_\phi(\boldsymbol{s}|\boldsymbol{x}) = \mathrm{Bern}(\boldsymbol{s}|\pi_\phi(\boldsymbol{x})) = \prod_{j=1}^p \pi_{\phi,j}(\boldsymbol{x})^{s_j}(1 - \pi_{\phi,j}(\boldsymbol{x}))^{1-s_j}. \tag{10}$$

Here $\pi_j$ is the estimated probability of being observed for that particular observation for feature $j$. The mapping $\pi_{\phi,j}(\boldsymbol{x})$ from the data to the probability of being observed for the $j$'th feature can be as general or specific as needed. A simple example could be that of *self-masking* or *self-censoring*, where the probability of the $j$'th feature being observed is only dependent on the feature value, $x_j$. Here the mapping can be a sigmoid on a linear mapping of the feature value, $\pi_{\phi,j}(\boldsymbol{x}) = \sigma(ax_j + b)$. The missing model can also be based on a group theoretic approach, see appendix C.

## 4 Experiments

In this section we apply the not-MIWAE to problems with values MNAR: censoring in multivariate datasets, clipping in images and selection bias in recommender systems. Implementation details and a link to source code can be found in appendix A.

| | Banknote | Concrete | Red | White | Yeast | Breast |
|---|---|---|---|---|---|---|
| PPCA | $1.39 \pm 0.00$ | $1.61 \pm 0.00$ | $1.61 \pm 0.00$ | $1.57 \pm 0.00$ | $1.67 \pm 0.00$ | $0.90 \pm 0.00$ |
| not-MIWAE - PPCA | | | | | | |
|   agnostic | $1.25 \pm 0.15$ | $1.47 \pm 0.01$ | $1.32 \pm 0.00$ | $1.27 \pm 0.01$ | $1.20 \pm 0.05$ | $0.78 \pm 0.00$ |
|   self-masking | $\mathbf{0.57 \pm 0.00}$ | $1.31 \pm 0.00$ | $\mathbf{1.13 \pm 0.00}$ | $\mathbf{0.99 \pm 0.00}$ | $0.78 \pm 0.00$ | $\mathbf{0.72 \pm 0.00}$ |
|   self-masking known | $\mathbf{0.57 \pm 0.00}$ | $\mathbf{1.31 \pm 0.00}$ | $\mathbf{1.13 \pm 0.00}$ | $\mathbf{0.99 \pm 0.00}$ | $0.77 \pm 0.00$ | $\mathbf{0.72 \pm 0.00}$ |
| MIWAE | $1.19 \pm 0.01$ | $1.66 \pm 0.01$ | $1.62 \pm 0.01$ | $1.55 \pm 0.01$ | $1.72 \pm 0.01$ | $1.20 \pm 0.01$ |
| not-MIWAE | | | | | | |
|   agnostic | $0.80 \pm 0.08$ | $2.63 \pm 0.12$ | $1.30 \pm 0.01$ | $1.37 \pm 0.00$ | $1.43 \pm 0.02$ | $1.10 \pm 0.01$ |
|   self-masking | $1.88 \pm 0.85$ | $1.26 \pm 0.02$ | $1.08 \pm 0.02$ | $1.04 \pm 0.01$ | $1.48 \pm 0.03$ | $\mathbf{0.74 \pm 0.01}$ |
|   self-masking known | $\mathbf{0.74 \pm 0.05}$ | $\mathbf{1.12 \pm 0.04}$ | $\mathbf{1.07 \pm 0.00}$ | $1.04 \pm 0.00$ | $1.38 \pm 0.02$ | $0.76 \pm 0.01$ |
| low-rank joint model | $\mathbf{0.79 \pm 0.02}$ | $\mathbf{1.57 \pm 0.01}$ | $\mathbf{1.42 \pm 0.01}$ | $\mathbf{1.39 \pm 0.01}$ | $\mathbf{1.19 \pm 0.00}$ | $1.22 \pm 0.01$ |
| missForest | $1.28 \pm 0.00$ | $1.76 \pm 0.01$ | $1.64 \pm 0.00$ | $1.63 \pm 0.00$ | $1.66 \pm 0.00$ | $1.57 \pm 0.00$ |
| MICE | $1.41 \pm 0.00$ | $1.70 \pm 0.00$ | $1.68 \pm 0.00$ | $1.41 \pm 0.00$ | $1.72 \pm 0.00$ | $\mathbf{1.17 \pm 0.00}$ |
| mean | $1.73 \pm 0.00$ | $1.85 \pm 0.00$ | $1.83 \pm 0.00$ | $1.74 \pm 0.00$ | $1.69 \pm 0.00$ | $1.82 \pm 0.00$ |

Table 1: Imputation RMSE on UCI datasets affecfed by MNAR.

## 4.1 EVALUATION METRICS

Model performance can be assessed using different metrics. A first metric would be to look at how well the marginal distribution of the data has been inferred. This can be assessed, if we happen to have a fully observed test-set available. Indeed, we can look at the test log-likelihood of this fully observed test-set as a measure of how close $p_\theta(\boldsymbol{x})$ and the true distribution of $\boldsymbol{x}$ are. In the case of a DLVM, performance can be estimated using importance sampling with the variational distribution as proposal (Rezende et al., 2014). Since the encoder is tuned to observations with missing data, it should be retrained (while keeping the decoder fixed) as suggested by Mattei & Frellsen (2018b).

Another metric of interest is the imputation error. In experimental settings where the missing mechanism is under our control, we have access to the actual values of the missing data and the imputation error can be found directly as an error measure between these and the reconstructions from the model. In real-world datasets affected by MNAR processes, we cannot use the usual approach of doing a train-test split of the observed data. As the test-set is biased by the same missing mechanism as the training-set it is not representative of the full population. Here we need a MAR data sample to evaluate model performance (Marlin et al., 2007).

## 4.2 SINGLE IMPUTATION IN UCI DATA SETS AFFECTED BY MNAR

We compare different imputation techniques on datasets from the UCI database (Dua & Graff, 2017), where in an MCAR setting the MIWAE has shown state of the art performance (Mattei & Frellsen, 2019). An MNAR missing process is introduced by self-masking in half of the features: when the feature value is higher than the feature mean it is set to missing. The MIWAE and not-MIWAE, as well as their linear PPCA-like versions, are fitted to the data with missing values. For the not-MIWAE three different approaches to the missing model are used: 1) *agnostic* where the data model output is mapped to logits for the missing process via a single dense linear layer, 2) *self-masking* where logistic regression is used for each feature and 3) *self-masking known* where the sign of the weights in the logistic regression is known.

We compare to the low-rank approximation of the concatenation of data and mask by Sportisse et al. (2020a) that is implicitly modelling the data and mask jointly. Furthermore we compare to mean imputation, missForest (Stekhoven & Bühlmann, 2012) and MICE (Buuren & Groothuis-Oudshoorn, 2010) using Bayesian Ridge regression. Similar settings are used for the MIWAE and not-MIWAE, see appendix A. Results over 5 runs are seen in table 1. Results for varying missing rates are in appendix E.

The low-rank joint model is almost always better than PPCA, missForest, MICE and mean, i.e. all M(C)AR approaches, which can be attributed to the implicit modelling of data and mask together. At the same time the not-MIWAE PPCA is always better than the corresponding low-rank joint model, except for the agnostic missing model on the Yeast dataset. Supplying the missing model with more knowledge of the missing process (that it is self-masking and the direction of the missing mechanism) improves performance. The not-MIWAE performance is also improved with more knowledge in the missing model. The agnostic missing process can give good performance, but is

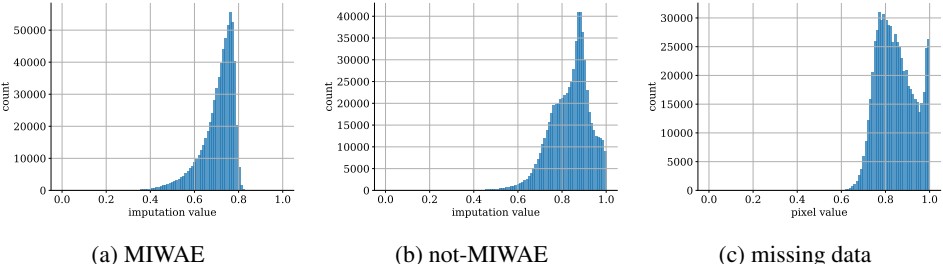

|(a) MIWAE|(b) not-MIWAE|(c) missing data|

Figure 2: SVHN: Histograms over imputed values for (a) the MIWAE and (b) the not-MIWAE, and (c) the pixel values of the missing data.

| Model | RMSE | $\mathcal{L}^{\text{test}}_{10000}$ |
|---|---|---|
| MIWAE | 0.17298 | 1867.66 |
| not-MIWAE | 0.07294 | 1894.36 |
| MIWAE no missing | | 1908.11 |

Figure 3: Rows from top: original images, images with missing, not-MIWAE imputations, MIWAE imputations

Table 2: SVHN: Imputation RMSE and test-set log-likelihood estimate. Constant imputation with 1's has a RMSE of 0.1757.

often led astray by an incorrectly learned missing model. This speaks to the trade-off between data model flexibility and missing model flexibility. The not-MIWAE PPCA has huge inductive bias in the data model and so we can employ a more flexible missing model and still get good results. For the not-MIWAE having both a flexible data model and a flexible missing model can be detrimental to performance. One way to asses the learnt missing processes is the mask classification accuracy on fully observed data. These are reported in table A1 and show that the accuracy increases as more information is put into the missing model.

### 4.3 CLIPPING IN SVHN IMAGES

We emulate the clipping phenomenon in images on the street view house numbers dataset (SVHN, Netzer et al., 2011). Here we introduce a self-masking missing mechanism that is identical for all pixels. The missing data is Bernoulli sampled with probability

$$\Pr(s_{ij} = 1 | x_{ij}) = \frac{1}{1 + e^{-\text{logits}}} \; , \; \text{logits} = W(x_{ij} - b), \tag{11}$$

where $W = -50$ and $b = 0.75$. This mimmicks a clipping process where $0.75$ is the clipping point (the data is converted to gray scale in the $[0, 1]$ range). For this experiment we use the true missing process as the missing model in the not-MIWAE.

Table 2 shows model performance in terms of imputation RMSE and test-set log likelihood as estimated with 10k importance samples. The not-MIWAE outperforms the MIWAE both in terms of test-set log likelihood and imputation RMSE. This is further illustrated in the imputations shown in figure 3. Since the MIWAE is only fitting the observed data, the range of pixel values in the imputations is limited compared to the true range. The not-MIWAE is forced to push some of the data-distribution towards higher pixel values, in order to get a higher likelihood in the logistic regression in the missing model. In figures 2a–2c, histograms over the imputation values are shown together with the true pixel values of the missing data. Here we see that the not-MIWAE puts a considerable amount of probability mass above the clipping value.

### 4.4 SELECTION BIAS IN THE YAHOO! R3 DATASET

The Yahoo! R3 dataset (webscope.sandbox.yahoo.com) contains ratings on a scale from 1–5 of songs in the database of the Yahoo! LaunchCast internet radio service and was first presented in (Marlin et al., 2007). It consists of two datasets with the same 1,000 songs selected randomly from

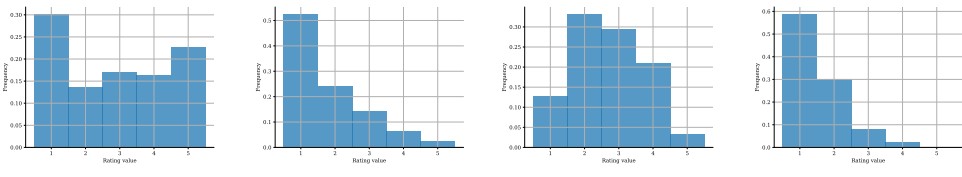

| (a) MNAR train samples | (b) MCAR test samples | (c) MIWAE impute | (d) not-MIWAE impute |

Figure 4: Histograms over rating values for the Yahoo! R3 dataset from (a) the MNAR training set and (b) the MCAR test set. (c) and (d) show histograms over imputations of missing values in the test set, when encoding the corresponding training set. The not-MIWAE imputations (d) are much more faithful to the shape of the test set (b) than the MIWAE imputations (c).

the LaunchCast database. The first dataset is considered an MNAR training set and contains self-selected ratings from 15,400 users. In the second dataset, considered an MCAR test-set, 5,400 of these users were asked to rate exactly 10 randomly selected songs. This gives a unique opportunity to train a model on a real-world MNAR-affected dataset while being able to get an unbiased estimate of the imputation error, due to the availability of MCAR ratings. The plausibility that the set of self-selected ratings was subject to an MNAR missing process was explored and substantiated by Marlin et al. (2007). The marginal distributions of samples from the self-selected dataset and the randomly selected dataset can be seen in figures 4a and 4b.

We train the MIWAE and the not-MIWAE on the MNAR ratings and evaluate the imputation error on the MCAR ratings. Both a gaussian and a categorical observation model is explored. In order to get reparameterized samples in the data space for the categorical observation model, we use the Gumbel-Softmax trick (Jang et al., 2017) with a temperature of $0.5$. The missing model is a logistic regression for each item/feature, with a shared weight across features and individual biases. A description of competitors can be found in appendix A.3 and follows the setup in (Wang et al., 2019). The results are grouped in table 3, from top to bottom, according to models not including the missing process (MAR approaches), models using propensity scoring techniques to debias training losses, and finally models learning a data model and a missing model jointly, without the use of propensity estimates.

The not-MIWAE shows state of the art performance, also compared to models based on propensity scores. The propensity based techniques need access to a small sample of MCAR data, i.e. a part of the test-set, to estimate the propensities using Naive Bayes, though they can be estimated using logistic regression if covariates are available (Schnabel et al., 2016) or using a nuclear-norm-constrained matrix factorization of the missing mask itself (Ma & Chen, 2019). *We stress that the not-MIWAE does not need access to similar unbiased data in order to learn the missing model.* However, the missing model in the not-MIWAE can take available information into account, e.g. we could fit a continuous mapping to the propensities and use this as the missing model, if propensities were available. Histograms over imputations for the missing data in the MCAR test-set can be seen for the MIWAE and not-MIWAE in figures 4c and 4d. The marginal distribution of the not-MIWAE imputations are seen to match that of the MCAR test-set better than the marginal distribution of the MIWAE imputations.

| Model | MSE |
|---|---|
| MF | 1.891 |
| PMF | 1.709 |
| AutoRec | 1.438 |
| Gaussian-VAE | 1.381 |
| MIWAE categorical | $2.067 \pm 0.004$ |
| MIWAE Gaussian | $2.055 \pm 0.001$ |
| CPT-v | 1.115 |
| MF-IPS | 0.989 |
| MF-DR-JL | 0.966 |
| NFM-DR-JL | 0.957 |
| MF-MNAR | 2.199 |
| Logit-vd | 1.301 |
| not-MIWAE categorical | $1.293 \pm 0.006$ |
| not-MIWAE gaussian | $\mathbf{0.939 \pm 0.007}$ |

Table 3: Imputation MSEs for the Yahoo! MCAR test-set. Models are trained on the MNAR training set.

## 5  CONCLUSION

The proposed not-MIWAE is versatile both in terms of defining missing mechanisms and in terms of application area. There is a trade-off between data model complexity and missing model complexity. In a parsimonious data model a very general missing process can be used while in flexible data

model the missing model needs to be more informative. Specifically, any knowledge about the missing process should be incorporated in the missing model to improve model performance. Doing so using recent advances in equivariant/invariant neural networks is an interesting avenue for future research (see appendix C). Recent developments on the subject of recoverability/identifiability of MNAR models (Sadinle & Reiter, 2018; Mohan & Pearl, 2021; Nabi et al., 2020; Sportisse et al., 2020b) could also be leveraged to design provably idenfiable not-MIWAE models.

Several extensions of the graphical models used here could be explored. For example, one could break off the conditional independence assumptions, in particular the one of the mask given the data. This could, for example, be done by using an additional latent variable pointing directly to the mask. Combined with a discriminative classifier, the not-MIWAE model could also be used in supervised learning with input values missing not at random following the techniques by Ipsen et al. (2020).

### ACKNOWLEDGMENTS

The Danish Innovation Foundation supported this work through Danish Center for Big Data Analytics driven Innovation (DABAI). JF acknowledge funding from the Independent Research Fund Denmark (grant number 9131-00082B) and the Novo Nordisk Foundation (grant numbers NNF20OC0062606 and NNF20OC0065611).

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

|  | Banknote | Concrete | Red | White | Yeast | Breast |
|---|---|---|---|---|---|---|
| not-MIWAE - PPCA | | | | | | |
| agnostic | $0.80 \pm 0.03$ | $0.75 \pm 0.05$ | $0.88 \pm 0.01$ | $0.83 \pm 0.00$ | $0.78 \pm 0.02$ | $0.96 \pm 0.00$ |
| self-masking | $0.92 \pm 0.05$ | $0.95 \pm 0.00$ | $0.96 \pm 0.00$ | $0.97 \pm 0.00$ | $0.99 \pm 0.00$ | $0.98 \pm 0.00$ |
| self-masking known | $0.98 \pm 0.00$ | $0.95 \pm 0.00$ | $0.96 \pm 0.00$ | $0.97 \pm 0.00$ | $1.00 \pm 0.00$ | $0.97 \pm 0.00$ |
| not-MIWAE | | | | | | |
| agnostic | $0.92 \pm 0.01$ | $0.54 \pm 0.04$ | $0.91 \pm 0.00$ | $0.88 \pm 0.00$ | $0.80 \pm 0.00$ | $0.93 \pm 0.00$ |
| self-masking | $0.99 \pm 0.00$ | $0.93 \pm 0.02$ | $0.95 \pm 0.01$ | $0.90 \pm 0.02$ | $0.71 \pm 0.02$ | $0.98 \pm 0.00$ |
| self-masking known | $0.99 \pm 0.00$ | $0.97 \pm 0.00$ | $0.97 \pm 0.00$ | $0.95 \pm 0.00$ | $0.78 \pm 0.00$ | $0.98 \pm 0.00$ |

Table A1: Mask prediction accuracies on UCI datasets using fully observed data.

## A    IMPLEMENTATION DETAILS

In all experiments we used TensorFlow probability (Dillon et al., 2017) and the Adam optimizer (Kingma & Ba, 2014) with a learning rate of 0.001. Gaussian distributions were used both as the variational distribution in latent space and the observation model in data space. No regularization was used. Similar settings were used for the MIWAE and the not-MIWAE, except for the missing model which is exclusive to the not-MIWAE.

Source code is available at: `https://github.com/nbip/notMIWAE`

### A.1    UCI

The encoder and decoder consist of two hidden layers with 128 units and `tanh` activation functions. In the PPCA-like models, the decoder is a linear mapping from latent space to data space, with a learnt variance shared across features. The size of the latent space is set to $p - 1$, $K = 20$ importance samples were used during training and a batch size of 16 was used for 100k iterations. Data are standardized before missing is introduced. The imputation RMSE is estimated using 10k importance samples and the mean and standard errors are found over 5 runs.

Since the imputation error in a real-world setting cannot be monitored during training, neither on a train or validation set, early stopping cannot be done based on this. Both the MIWAE and not-MIWAE are trained for a fixed number of iterations. In the low-rank joint model of Sportisse et al. (2020a), model selection needs to be done for the penalization parameter $\lambda$[1]. In order to do this we add 5% missing values (MCAR) to the concatenated matrix of data and mask and use the imputation error on this added missing data to select the optimal lambda. The model is then trained on the original data using the optimal $\lambda$ to get the imputation error.

For evaluating the learnt missing model, we report mask classification accuracies when feeding fully observed data as input to the missing model, see table A1. As the missing model contains more prior information, the classification accuracy becomes better and better.

### A.2    SVHN

For the encoder and decoder a convolutional structure was used (see tables A2 and A3) together with ReLU activations and a latent space of dimension 20. $K = 5$ importance samples were used during training and a batch size of 64 was used for 1M iterations. The variance in the observation model was lower bounded at $\sim 0.02$.

### A.3    YAHOO!

The MIWAE and the not-MIWAE were trained on the MNAR ratings and the imputation error was evaluated on the MCAR ratings (when encoding the MNAR ratings). We used the permutation invariant encoder by Ma et al. (2018) with an embedding size of 20 and a code size of 50, along with a linear mapping to a latent space of size 30. In the Gaussian observation model, the decoder is a linear mapping and there is a sigmoid activation of the mean in data space, scaled to match the scale

---

[1]We used original code from the authors found here: `https://github.com/AudeSportisse/stat`

Table A2: SVHN encoder

| layer(size) |
| --- |
| Input $x$ ($32 \times 32 \times 1$) |
| Conv2D($16 \times 16 \times 64$) |
| Conv2D($8 \times 8 \times 128$) |
| Conv2D($4 \times 4 \times 256$) |
| Reshape(4096) |
| $\mu$: Dense(20) |
| $\log \sigma$: Dense(20) |

Table A3: SVHN decoder

| layer(size) |
| --- |
| Latent variable $z$(20) |
| Dense(4096) |
| Reshape($4 \times 4 \times 256$) |
| Conv2Dtranspose($8 \times 8 \times 256$) |
| Conv2Dtranspose($16 \times 16 \times 128$) |
| $\mu$:
Conv2Dtranspose($32 \times 32 \times 64$)
Conv2Dtranspose($32 \times 32 \times 1$)
sigmoid |
| $\log \sigma$:
Conv2Dtranspose($32 \times 32 \times 64$)
Conv2Dtranspose($32 \times 32 \times 1$) |

of the ratings. The categorical observation model also has a linear mapping to its logits. In both latent space and data space, we learn shared variance parameters in each dimension. The missing model is a logistic regression for each feature, with a shared weight across features and individual biases for each feature. We use $K = 20$ importance samples during training, ReLU activations, a batch size of 100 and train for 10k iterations.

We follow the setup of Wang et al. (2019) and compare to the following approaches:

**CPT-v**: Marlin et al. (2007) show that a multinomial mixture model with a Conditional Probability Tables missing model give better performance than the multinomial mixture model without missing model. The approach is further expanded by Marlin & Zemel (2009), where a logistic model, **Logit-vd**, is also tried as the missing model. The result for the CPT-v model and the Logit-vd model are taken from the supplementary material of Hernández-Lobato et al. (2014).

**MF-MNAR**: Hernández-Lobato et al. (2014) extended probabilistic matrix factorization to include a missing data model for data missing not at random in a collaborative filtering setting. Results are from the supplementary material of the paper.

**MF-IPS**: Schnabel et al. (2016) applied propensity-based methods from causal inference to matrix factorization, specifically *inverse-propensity-scoring, IPS*. The propensities used to debias the matrix factorization are the probabilities of a rating being observed for each (user, item) pair. The propensities used for training are found using 5% of the MCAR test-set. Results are from the paper.

**MF-DR-JL** and **NFM-DR-JL**: Wang et al. (2019) combines the propensity-scoring approach from Schnabel et al. (2016) with an error-imputation approach by Steck (2013) to obtain a doubly robust estimator. This is used both with matrix factorization and in *neural factorization machines* (He & Chua, 2017). As for Schnabel et al. (2016), 5% of the MCAR test-set is used to learn the propensities. Results are from the paper.

In addition to these debiasing approaches, we compare to the following methods, which do not take the missing process into account: **MF** (Koren et al., 2009), **PMF** (Mnih & Salakhutdinov, 2008), **AutoRec** (Sedhain et al., 2015) and **Gaussian VAE** (Liang et al., 2018). The presented results for these methods are from (Wang et al., 2019).

## B IMPUTATION

Once the model has been trained, it is possible to use it to impute the missing values. If our performance metric is a loss function $L(\boldsymbol{x}^{\mathrm{m}}, \boldsymbol{y}^{\mathrm{m}})$, optimal imputations $\hat{\boldsymbol{x}}^{\mathrm{m}}$ minimise $\mathbb{E}_{\boldsymbol{x}^{\mathrm{m}}}[L(\boldsymbol{x}^{\mathrm{m}}, \hat{\boldsymbol{x}}^{\mathrm{m}}) | \boldsymbol{x}^{\mathrm{o}}, \boldsymbol{s}]$. Many loss functions can be minimized using moments of the conditional distribution of the missing values, given the observed. Similarly to Mattei & Frellsen (2019, equations 10–11), these moments can be estimated via self-normalised importance sampling. For any

function of the missing data $h(\boldsymbol{x}^{\mathrm{m}})$,

$$\mathbb{E}[h(\boldsymbol{x}^{\mathrm{m}})|\boldsymbol{x}^{\mathrm{o}}, \boldsymbol{s}] = \int h(\boldsymbol{x}^{\mathrm{m}})p(\boldsymbol{x}^{\mathrm{m}}|\boldsymbol{x}^{\mathrm{o}}, \boldsymbol{s}) \, \mathrm{d}\boldsymbol{x}^{\mathrm{m}}. \tag{12}$$

Using Bayes's theorem, we get

$$\mathbb{E}[h(\boldsymbol{x}^{\mathrm{m}})|\boldsymbol{x}^{\mathrm{o}}, \boldsymbol{s}] = \int h(\boldsymbol{x}^{\mathrm{m}})\frac{p(\boldsymbol{s}|\boldsymbol{x}^{\mathrm{o}}, \boldsymbol{x}^{\mathrm{m}})p(\boldsymbol{x}^{\mathrm{m}}, \boldsymbol{x}^{\mathrm{o}})}{p(\boldsymbol{s}, \boldsymbol{x}^{\mathrm{o}})} \, \mathrm{d}\boldsymbol{x}^{\mathrm{m}}, \tag{13}$$

and now we can introduce the latent variable:

$$\mathbb{E}[h(\boldsymbol{x}^{\mathrm{m}})|\boldsymbol{x}_i^{\mathrm{o}}, \boldsymbol{s}] = \iint h(\boldsymbol{x}^{\mathrm{m}})\frac{p(\boldsymbol{s}|\boldsymbol{x}^{\mathrm{o}}, \boldsymbol{x}^{\mathrm{m}})p(\boldsymbol{x}^{\mathrm{m}}|\boldsymbol{z})p(\boldsymbol{x}^{\mathrm{o}}|\boldsymbol{z})p(\boldsymbol{z})}{p(\boldsymbol{s}, \boldsymbol{x}^{\mathrm{o}})} \, \mathrm{d}\boldsymbol{z} \, \mathrm{d}\boldsymbol{x}^{\mathrm{m}}. \tag{14}$$

Using self-normalised importance sampling on this last integral with proposal $q_\gamma(\boldsymbol{z}|\boldsymbol{x}^{\mathrm{o}})p_\theta(\boldsymbol{x}^{\mathrm{m}}|\boldsymbol{z})$ leads to the estimate

$$\hat{\boldsymbol{x}}^{\mathrm{m}} = \mathbb{E}[h(\boldsymbol{x}^{\mathrm{m}})|\boldsymbol{x}^{\mathrm{o}}, \boldsymbol{s}] \approx \sum_{k=1}^{K} \alpha_k h(\boldsymbol{x}_k^{\mathrm{m}}), \text{ with } \alpha_k = \frac{w_k}{w_1 + \ldots + w_K}, \tag{15}$$

where the weights $w_1, \ldots, w_K$ are incidentally identical to the ones used for training:

$$\forall k \leq K, \; w_k = \frac{p_\phi(\boldsymbol{s}|\boldsymbol{x}^{\mathrm{o}}, \boldsymbol{x}_k^{\mathrm{m}})p_\theta(\boldsymbol{x}^{\mathrm{o}}|\boldsymbol{z}_k)p(\boldsymbol{z}_k)}{q_\gamma(\boldsymbol{z}_k|\boldsymbol{x}^{\mathrm{o}})}, \tag{16}$$

and $(\boldsymbol{z}_1, \boldsymbol{x}_1^{\mathrm{m}}), \ldots, (\boldsymbol{z}_K, \boldsymbol{x}_K^{\mathrm{m}})$ are $K$ i.i.d. samples from $q_\gamma(\boldsymbol{z}|\boldsymbol{x}^{\mathrm{o}})p_\theta(\boldsymbol{x}^{\mathrm{m}}|\boldsymbol{z})$. If the quantity $\mathbb{E}[h(\boldsymbol{x}^{\mathrm{m}})|\boldsymbol{z}]$ is easy to compute, then a Rao-Blackwellized version of equation (15) should be preferred

$$\hat{\boldsymbol{x}}^{\mathrm{m}} = \mathbb{E}[h(\boldsymbol{x}^{\mathrm{m}})|\boldsymbol{x}^{\mathrm{o}}, \boldsymbol{s}] \approx \sum_{k=1}^{K} \alpha_k \mathbb{E}[h(\boldsymbol{x}^{\mathrm{m}})|\boldsymbol{z}_k]. \tag{17}$$

**Squared loss** When $L$ corresponds to the squared error, the optimal imputation will be the conditional mean that can be estimated using the method above (in that case, $h$ is the identity function):

$$\hat{\boldsymbol{x}}^{\mathrm{m}} = \mathbb{E}[\boldsymbol{x}^{\mathrm{m}}|\boldsymbol{x}^{\mathrm{o}}, \boldsymbol{s}] \approx \sum_{k=1}^{K} \alpha_k \mathbb{E}[\boldsymbol{x}^{\mathrm{m}}|\boldsymbol{x}^{\mathrm{o}}, \boldsymbol{s}], \text{ with } \alpha_k = \frac{w_k}{w_1 + \ldots + w_K}. \tag{18}$$

**Absolute loss** When $L$ is the absolute error loss, the optimal imputation is the conditional median, that can be estimated using the same technique and at little additional cost compared to the mean. Indeed, we can estimate the cumulative distribution function of each missing feature $j \in \{1, \ldots, p\}$:

$$F_j(x_j) = \mathbb{E}[\mathbf{1}_{x_j^{\mathrm{m}} \leq x_j}|\boldsymbol{x}^{\mathrm{o}}, \boldsymbol{s}] \approx \sum_{k=1}^{K} \alpha_k F_{x_j|\boldsymbol{x}^{\mathrm{o}}, \boldsymbol{s}}(x_j), \tag{19}$$

where $F_{x_j|\boldsymbol{x}^{\mathrm{o}}, \boldsymbol{s}}$ is the cumulative distribution function of $x_j|\boldsymbol{x}^{\mathrm{o}}, \boldsymbol{s}$, which will often be available in closed-form (e.g. in the case of a Gaussian, Bernoulli or Student's $t$ observation model). We can then use this estimate to approximately solve $F_j(x_j) = 0.5$. More generally, if $L$ is a multilinear loss, optimal imputations will be quantiles (see e.g. Robert, 2007, section 2.5.2) that can be estimated using equation (19). The consistency of similar quantile estimates was studied by Glynn (1996).

**Multiple imputation.** It is also possible to perform multiple imputation with the same computations. One can obtain approximate samples from $p(\boldsymbol{x}^{\mathrm{m}}|\boldsymbol{x}^{\mathrm{o}})$ using sampling importance resampling with the same set of weights. This allows us to do both single and multiple imputation with the same computations.

## C    MISSING MODEL, GROUP THEORETIC APPROACH

A more complex form of prior information that can be used to choose the form of $\pi_\phi(\boldsymbol{x})$ is group-theoretic. For example, we may know a priori that $p_\phi(\boldsymbol{s}|\boldsymbol{x})$ is invariant to a certain group action $g \cdot \boldsymbol{x}$ on the data space:

$$\forall g, \; p_\phi(\boldsymbol{s}|\boldsymbol{x}) = p_\phi(\boldsymbol{s}|g \cdot \boldsymbol{x}). \tag{20}$$

This would for example be the case, if the data sets were made of images whose class is invariant to translations (which is the case of most image data sets, like MNIST or SVHN), and with a missing model only dependent on the class. Similarly, one may know that the missing process is equivariant:

$$\forall g, \ p_\phi(g \cdot \boldsymbol{s}|\boldsymbol{x}) = p_\phi(\boldsymbol{s}|g^{-1} \cdot \boldsymbol{x}). \tag{21}$$

Again, such a setting can appear when there is strong geometric structure in the data (e.g. with images or proteins). Invariance or equivariance can be built in the architecture of $\pi_\phi(\boldsymbol{x})$ by leveraging the quite large body of work on invariant/equivariant convolutional neural networks, see e.g. Bietti & Mairal (2017); Cohen et al. (2019); Zaheer et al. (2017); Wiqvist et al. (2019); Bloem-Reddy & Teh (2020), and references therein.

## D  THEORETICAL PROPERTIES OF THE NOT-MIWAE BOUND

The properties of the not-MIWAE bound are directly inherited from the ones of the usual IWAE bound. Indeed, as we will see, the not-MIWAE bound is a particular instance of IWAE bound with an extended latent space composed of both the code and the missing values. More specifically, recall the definition of the not-MIWAE bound

$$\mathcal{L}_K(\theta, \phi, \gamma) = \sum_{i=1}^n \mathbb{E}\left[\log \frac{1}{K} \sum_{k=1}^K w_{ki}\right], \ \ \text{with } w_{ki} = \frac{p_\theta(\boldsymbol{x}_i^{\text{o}}|\boldsymbol{z}_{ki})p_\phi(\boldsymbol{s}_i|\boldsymbol{x}_i^{\text{o}}, \boldsymbol{x}_{ki}^{\text{m}})p(\boldsymbol{z}_{ki})}{q_\gamma(\boldsymbol{z}_{ki}|\boldsymbol{x}_i^{\text{o}})}. \tag{22}$$

Each $i$th term of the sum can be seen as an IWAE bound with extended latent variable $(\boldsymbol{z}_{ki}, \boldsymbol{x}_{ki}^{\text{m}})$, whose prior is $p_\theta(\boldsymbol{x}_{ki}^{\text{m}}|\boldsymbol{z}_{ki})p(\boldsymbol{z}_{ki})$. The related importance sampling proposal of the $i$th term is $p_\theta(\boldsymbol{x}_{ki}^{\text{m}}|\boldsymbol{z}_{ki})q_\gamma(\boldsymbol{z}_{ki}|\boldsymbol{x}_i^{\text{o}})$, and the observation model is $p_\phi(\boldsymbol{s}_i|\boldsymbol{x}_i^{\text{o}}, \boldsymbol{x}_{ki}^{\text{m}})p_\theta(\boldsymbol{x}_i^{\text{o}}|\boldsymbol{z}_{ki})$.

Since all $n$ terms of the sum are IWAE bounds, Theorem 1 from Burda et al. (2016) directly gives the monotonicity property:

$$\mathcal{L}_1(\theta, \phi, \gamma) \leq \ldots \leq \mathcal{L}_K(\theta, \phi, \gamma). \tag{23}$$

Regarding convergence of the bound to the true likelihood, we can use Theorem 3 of Domke & Sheldon (2018) for each term of the sum to get the following result.

**Theorem.** *Assuming that, for all $i \in \{1, ..., n\}$,*

- *there exists $\alpha_i > 0$ such that $\mathbb{E}\left[|w_{1i} - p_{\theta,\phi}(\boldsymbol{x}_i^o, \boldsymbol{s}_i)|^{2+\alpha_i}\right] < \infty$,*

- $\limsup_{K \longrightarrow \infty} \mathbb{E}\left[K/(w_{1i} + ... + w_{Ki})\right] < \infty$,

*the not-MIWAE bound converges to the true likelihood at rate $1/K$:*

$$\ell(\theta, \phi) - \mathcal{L}_K(\theta, \phi, \gamma) \underset{K \to \infty}{\sim} \frac{1}{K} \sum_{i=1}^n \frac{\text{Var}[w_{1i}]}{2p_{\theta,\phi}(\boldsymbol{x}_i^o, \boldsymbol{s}_i)^2}. \tag{24}$$

## E  VARYING MISSING RATE (UCI)

The UCI experiments use a self-masking missing process in half the features: when the feature value is higher than the feature mean it is set to missing. In order to investigate varying missing rates we change the cutoff point from the mean to **the mean plus an offset**. The offsets used are $\{0, 0.25, 0.5, 0.75, 1.0\}$, so that the largest cutoff point will be the mean plus one standard deviation. Increasing the cutoff point further results in mainly imputing outliers. Results for PPCA and not-MIWAE PPCA using the agnostic missing model are seen in figure 5 and using the self-masking model with known sign of the weights are seen in figure 6. Figure 7 shows the results for MIWAE and not-MIWAE using self-masking with known sign of the weights.

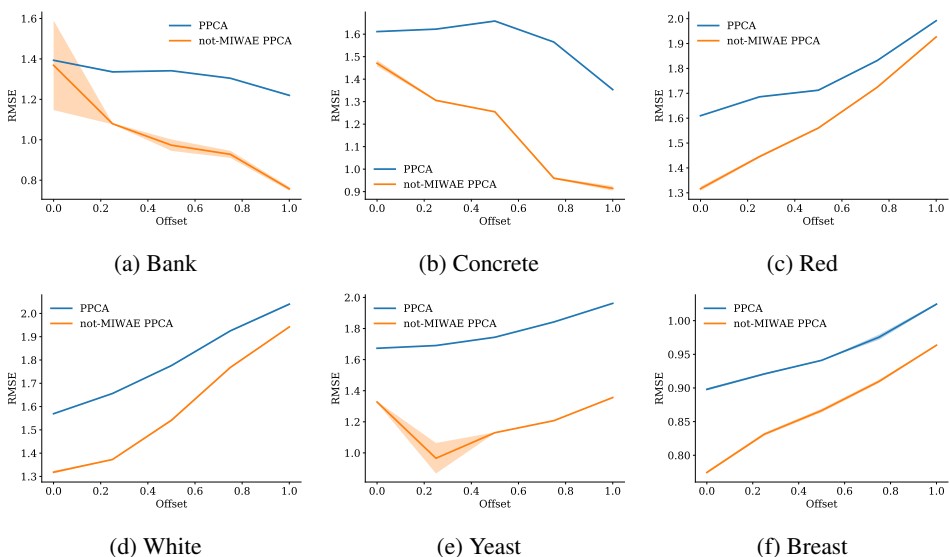

Figure 5: **PPCA agnostic**: Imputation RMSE at varying missing rates on UCI datasets. The variation in missing rate is obtained by changing the cutoff point using an offset, so that an offset = 0 corresponds to using the mean as the cutoff point while an offset = 1 corresponds to using the mean plus one standard deviation as the cutoff point. Results are averages over 2 runs.

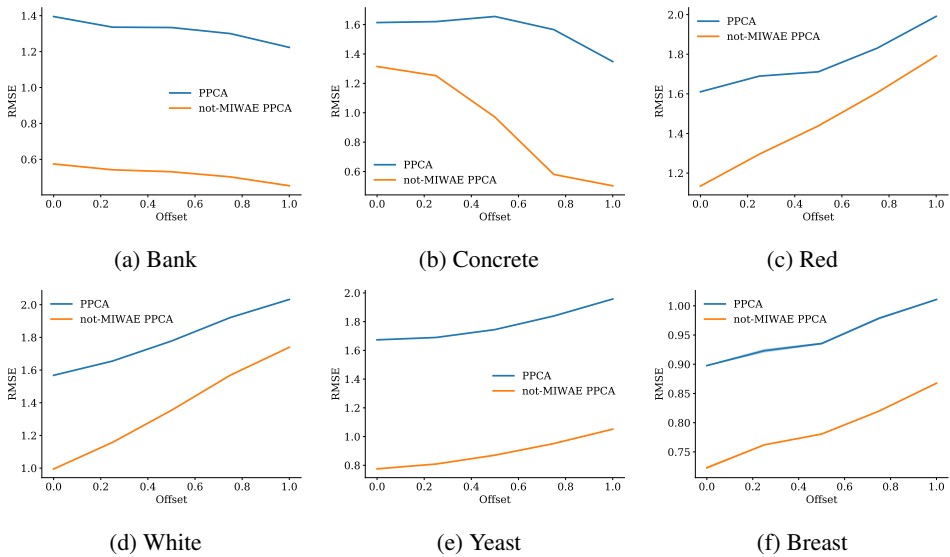

Figure 6: **PPCA self-masking known**: Imputation RMSE at varying missing rates on UCI datasets. The variation in missing rate is obtained by changing the cutoff point using an offset, so that an offset = 0 corresponds to using the mean as the cutoff point while an offset = 1 corresponds to using the mean plus one standard deviation as the cutoff point. Results are averages over 2 runs.

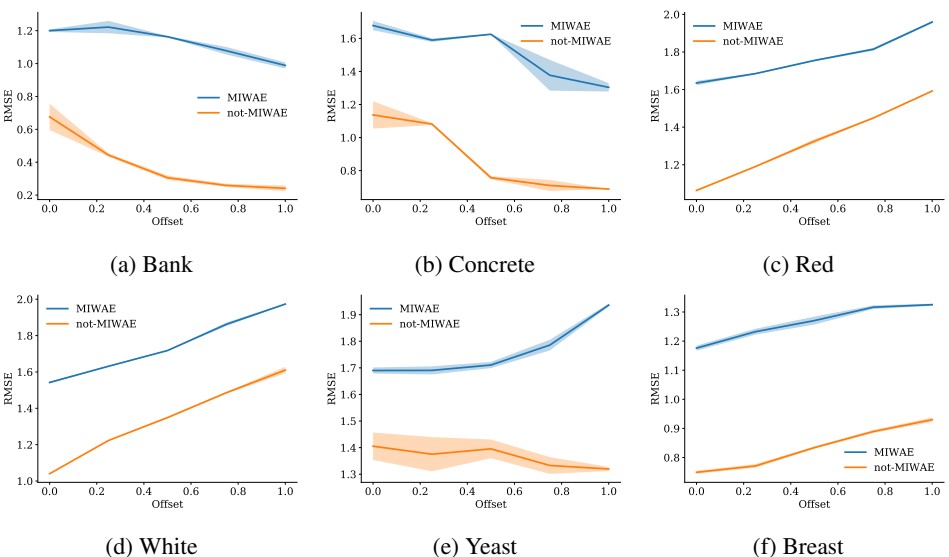

(a) Bank            (b) Concrete            (c) Red

(d) White            (e) Yeast            (f) Breast

Figure 7: **Self-masking known**: Imputation RMSE at varying missing rates on UCI datasets. The variation in missing rate is obtained by changing the cutoff point using an offset, so that an offset $= 0$ corresponds to using the mean as the cutoff point while an offset $= 1$ corresponds to using the mean plus one standard deviation as the cutoff point. Results are averages over 2 runs.

