# OpenReview forum: "not-MIWAE: Deep Generative Modelling with Missing not at Random Data"
_ICLR.cc/2021/Conference — ICLR 2021 Poster_

### Official Review · AnonReviewer4 · 2020-10-25
**The way that this paper adapts deep latent variable models for missing not at random data is quite straightforword and lacks technical depth.**

**Rating:** 4
**Confidence:** 3

**Review:**

This paper proposes an approach to training deep latent variable models on data that is missing not at random. To learn the parameters of deep latent variable models, the paper adopts importance-weighted variational inference techniques. Experiments on a variety of datasets show that the proposed approach is effective by explicitly modeling missing not at random data.

P1: The related work is well done. This paper reviews the most related studies in several lines of research, including missing data concepts and theories in statistics, missing not at random data in various applications, deep latent variable models for missing data, etc.

P2: The experimental results are quite extensive. This paper conducts experiments on a wide range of datasets from different domains: censoring on multi-variate datasets, clipping on image datasets, and bias on recommendation datasets. The paper also compares the proposed approach against a representative selection of state-of-the-art approaches.

C1: The main concern for this paper is the lack of technical depth. Using variational distribution to derive a tractable lower bound of of a joint likelihood function, using Monte Carlo estimates to unbiasedly approximate the lower bound, and using a reparameterization trick to obtain unbiased estimates of gradients of the lower bound are well-established techniques. It is important for this paper to highlight what is the novelty of the proposed approach in terms of technical innovation.

C2: This paper argues that the proposed approach allows for incorporating prior information about types of missingness. However, it is not clear to me what is the prior information and how does the proposed approach leverage the prior information. It is highly recommended for this paper to provide a formal formulation of the prior information in missing not at random data. It is also recommended for the paper to elaborate how the proposed approach uses the prior information and underlying motivation.

C3: The last sentence in the 4th page of this paper states that it is possible to show that a sequence of objective functions converges to the joint likelihood function. To make the statement more convincing, it would be better if the paper could include a proof that the sequence of objective functions theoretically converges.

---

> ### Author Response · Authors · 2020-11-19
> **RE: The way that this paper adapts deep latent variable models for missing not at random data is quite straightforword and lacks technical depth**
>
> Many thanks for your comments and assessment of our paper!
>
> We address your three points of critique below.
>
> C1: Our main contribution is to provide a general recipe for building and training MNAR models that leverages both the flexibility of deep neural nets and prior information about the missingness process. Regarding the technical novelty of the inference scheme, we believe that our use of the reparametrisation trick both in data space and code space is new, although it is a combination of popular generative modelling tools.
>
> C2: Thanks for pointing out that lack of clarity. Within not-MIWAE, we can use prior information by specifying the parametric form of $p(\mathbf{s}|\mathbf{x})$, as we explain in Section 3.1.
>
> Take for example the case of a sensor which has a tendency to fail at high levels. This can be incorporated in the structure of the missing model by modelling $p(\mathbf{s}|\mathbf{x})$ as a logistic regression for this specific feature, taking the sensor values as input while trying to predict the corresponding mask values. This is illustrated in the UCI experiments where three levels of knowledge are compared:
> 1. agnostic, where a fully connected neural network is used as the missing model
> 2. self-masking, where a logistic regression for each feature is used
> 3. self-masking known, where a logistic regression for each feature is used and the direction of the sigmoid is supplied as well
>
> To make things slightly clearer, we have renamed Section 3.1: “Using prior information via the missing data model”. Do you find the discussion in Section 3.1 clear enough?
>
> C3: Thanks for pointing out the vagueness of this theoretical paragraph. We agree that giving a precise result is more compelling, and added a new appendix (Appendix D), for that purpose in the revision. We explain why the theoretical properties of the not-MIWAE are directly inherited from those of the IWAE, and give the regularity conditions for convergence of the bound at rate $1/K$, following Domke and Sheldon (NeurIPS 2018).

---

### Official Review · AnonReviewer3 · 2020-10-26
**Relevant contribution to MNAR data imputation with potential for improvement in experimental validation**

**Rating:** 6
**Confidence:** 3

**Review:**

Overall I very much enjoyed reading this paper! The manuscript is very well written, the related work section is sound, the motivation is clear, the methodology is well formalized and the experimental validation is strong in some aspects.

The formalisation of the missingness process is very intuitive. Also the literature referenced provides one of the most comprehensive overviews on the topic in the statistics community, on the deep learning side it seems that there is a lot of research covered on the side of deep learning latent variable models based on variational auto encoders (VAEs), but the complementary work on Generative Adversarial Networks (GANs) appears to be less well covered. I’m myself more familiar with the VAE approach, and the authors correctly mention the implicit assumptions of GAN approaches, but as GANs offer an intuitive parametric and flexible way of modelling the missingness mask, I guess it would be helpful to see them in the comparison.

In the first paragraph on page 5 the authors mention that the approach with the reparametrization only works if the data is continuous. I might be missing something, but rather than using a sampling based approach as in Mohamed et al, referenced by the authors, maybe it would be an option to use something like the Gumbel Softmax? In the experimental section the authors use that approach for the recommender system data, with limited success, it seems, compared to a plain gaussian likelihood. But conceptually it would be good to comment on why that’s not possible?

My main concerns with this work is the experimental validation. The experimental settings explored are UCI data sets, image data and recommendation systems. It’s great that the authors provide such a comprehensive and heterogenous experimental validation in terms of data sets. What I found a bit limiting is that the not-missing-at-random process was so simple and restricted, and that the missingness ratio was not explored systematically. Many studies on imputation use experimental validation that explores missingness ratios between 0 and 100% of the values and in particular the MAR and MNAR settings are explored by, e.g. sampling a random quantile of a feature to condition the missingness on. This is very simple to implement but would allow for a much more realistic account of missingness structure. Especially for a study like this, which makes a presumably important and strong contribution to the field of missing data imputation I would recommend to demonstrate the effectiveness of the proposed approach by a more realistic experimental setting.

Another recommendation would be to highlight the advantage of the proposed approach with more synthetic data experiments as in figure 1b, maybe one linear and one non-linear data manifold. That would allow to control for more parameters like the rank of the data and the noise, their covariance structure (strength and independence of features and noise, respectively). But that’s not really necessary I think, the authors did a great job with figure 1b!

---

> ### Author Response · Authors · 2020-11-19
> **RE: Relevant contribution to MNAR data imputation with potential for improvement in experimental validation**
>
> Many thanks for your comments and assessment of our paper!
>
> > I’m myself more familiar with the VAE approach, and the authors correctly mention the implicit assumptions of GAN approaches, but as GANs offer an intuitive parametric and flexible way of modelling the missingness mask, I guess it would be helpful to see them in the comparison.
>
> Thank you for your positive comments on our literature review. It is unclear to us, if you are asking for a more thorough discussion of the literature for GAN with missing data, or you are asking for an empirical comparison to GAIN? The reason why we have not done an empirical comparison with GAIN, is because (1) this model assumes MCAR, and (2) Ivanov et al. (2019) found that GAIN did not significantly outperform missForest, which we compare to.
>
> > In the first paragraph on page 5 the authors mention that the approach with the reparametrization only works if the data is continuous. I might be missing something, but rather than using a sampling based approach as in Mohamed et al, referenced by the authors, maybe it would be an option to use something like the Gumbel Softmax?
>
> Yes, it is definitely an option to use the Gumbel Softmax (to make it clearer in the text, we have added citations to the Gumbel-softmax trick next to the citation of the review paper of Mohamed et al.). As to why it has less success in the Yahoo experiment, this is still unclear to us. This might be related to the reparameterization (since Gumbel-softmax induces a gradient bias), but also to the nature of the observation model compared to the data at hand.
>
> > What I found a bit limiting is that the not-missing-at-random process was so simple and restricted, and that the missingness ratio was not explored systematically. Many studies on imputation use experimental validation that explores missingness ratios between 0 and 100% of the values and in particular the MAR and MNAR settings are explored [...]
>
> Thank you for this comment. We would like to point out that the missing process for the Yahoo R3 data set is an example of a real world unknown missing process. A great avenue of future research would be to play with complex-but-not-agnostic missing processes like the ones we describe in the conclusion and in Appendix C. For complex missing processes, the question of recoverability is also quite essential: is the model simple enough to allow us to make correct inferences given enough data?
>
> In appendix E, we have added experiments for varying experimental results for missing rates in the UCI data set, where we compare the performance of PPCA and not-MIWAE PPCA using the agnostic missing model (the plots are currently done with two repetitions, but we will add more in the final version).
>
>
> Ivanov, O., Figurnov, M., and Vetrov, D. Variational autoencoder with arbitrary conditioning. In International Conference on Learning Representations, 2019.

---

> > ### Comment · AnonReviewer3 · 2020-11-19
> > **Additional experiments are a significant improvement**
> >
> > thanks for your clarifications, especially that one reference by Ivanov was helpful I wasn't aware of that.
> >
> > It makes sense to leave out GAIN or GAN approaches in the comparisons, given those results.
> >
> > Also, thanks for the additional experiments!

---

### Official Review · AnonReviewer2 · 2020-10-28
**Interesting contribution to handle missing data not at random**

**Rating:** 7
**Confidence:** 4

**Review:**

Review:

This paper handles the problem of missing not-at-random (MNAR) by extending the MIWAE model to MNAR scenarios. To do this, they use the reparametrization trick in the data space to get the stochastic gradients of the lower bound.

Minor questions/comments:

- At the beginning of section 2, $\mathbf{x}_i$ is defined as a row the data matrix. However, I have noticed that the $i$ subscript is dropped several times across the text and left simply as $\mathbf{x}$. Maybe a small comment indicating this at the beginning could help to not get confused while reading section 2.

- In section 4.4. the authors say that both a categorical and a Gaussian observation model are used. I was under the impression that this paper was only evaluated on real attributes, mainly based on the use of MSE and RMSE as evaluation metrics. Are the datasets in Table 1 also a mixture of discrete and continuous variables? In that case, how do the authors compute the MSE of the discrete variables in Table 3 and, if it applies, in Table 1?

Summary:

The problem of MNAR is very important in practical scenarios, specially when handling tabular data. The paper was well written and explained, and although it can be seen as a simple extension to the MIWAE model, nonetheless I consider it to be a relevant and interesting contribution.

---

> ### Author Response · Authors · 2020-11-19
> **RE: Interesting contribution to handle missing data not at random**
>
> Many thanks for your comments and assessment of our paper!
>
> > At the beginning of section 2, $\mathbf{x}_i$  is defined as a row the data matrix. However, I have noticed that the $i$ subscript is dropped several times across the text and left simply as $\mathbf{x}$. Maybe a small comment indicating this at the beginning could help to not get confused while reading section 2.
>
> Thank you, we have tried to clear that up by slightly modifying the beginning of Section 2 in the revision, and adding the sentence: "Throughout the text, we will make statements about the random variable $\mathbf{x}$, and only consider samples $\mathbf{x}_i$ when necessary."
>
> > In section 4.4. the authors say that both a categorical and a Gaussian observation model are used. I was under the impression that this paper was only evaluated on real attributes, mainly based on the use of MSE and RMSE as evaluation metrics. Are the datasets in Table 1 also a mixture of discrete and continuous variables? In that case, how do the authors compute the MSE of the discrete variables in Table 3 and, if it applies, in Table 1?
>
> The rating data {1,2,3,4,5} can be modelled as either continuous data or categorical data. When modelling it as categorical data you can still produce imputations that are not discrete since the mean of a categorical distribution is not discrete. The data are imputed using the approximate conditional mean $E [\mathbf{x}^\mathrm{m} | \mathbf{x}^\mathrm{o}, \mathbf{s}]$ computed with the techniques of Appendix B, and then the RMSE is computed. In table 1 there are some features that are semi-discrete as in integer values in some range. Binary features have been excluded, and a Gaussian observation model is used for all features.

---

### Official Review · AnonReviewer1 · 2020-10-28
**A generic VAE model for MNAR data**

**Rating:** 6
**Confidence:** 4

**Review:**

The paper introduces a deep latent variable model (DLVM) for missing data problems where the missing mechanism is missing not at random (MNAR) and therefore cannot be ignored. It presents an approach for fitting the model based on importance-weighted variational inference and reparameterization trick, and demonstrates the application of the proposed method in simulated and real data sets.

Pros:

-The paper addresses an important problem of dealing with MNAR data, by introducing a DLVM model that allows for incorporating prior information about the type of missingness (for example, self centoring) into the model. This extends the applicability of DLVM models to a wider class of practical problems.

-The paper is clearly written.

-The experimental results demonstrate the trade-off between data model flexibility and missing model flexibility.


Cons:

-The proposed approach is a relatively straightforward extension of the existing work (MIWAE, Mattei & Frellsen, 2019), using somewhat standard VAE techniques.

-I’m not sure the experimental results really demonstrated the advantages of the proposed methods against existing ones, particularly in the settings of no prior knowledge about the type of missingness mechanism. Though seeing examples of the trade-off between data model flexibility and missing model flexibility is nice.

Overall, I think the results in the paper should be useful in a number of applications and the paper has enough contributions to merit publication.

---

> ### Author Response · Authors · 2020-11-19
> **RE: A generic VAE model for MNAR data**
>
> Many thanks for your comments and assessment of our paper!
>
> > I’m not sure the experimental results really demonstrated the advantages of the proposed methods against existing ones, particularly in the settings of no prior knowledge about the type of missingness mechanism. Though seeing examples of the trade-off between data model flexibility and missing model flexibility is nice.
>
> We agree that, when there is no prior knowledge, the not-MIWAE is of limited interest. However,  we also believe that good MNAR modelling is mostly possible when there is prior knowledge. This assessment is consistent with the findings of several of the papers that we cite, in particular Molenberghs et al. (JRSSB 2008, beginning of Section 6).

---

### Author Response · Authors · 2020-11-19
**General response from authors**

We would like to thank the reviewers for the valuable feedback and we very much appreciate their assessment that “The paper addresses an important problem of dealing with MNAR data” and that “The paper is clearly written” / “The paper was well written and explained,” / “The manuscript is very well written” and that “Experiments on a variety of datasets show that the proposed approach is effective by explicitly modeling missing not at random data.”

Following your remarks, we have made the following modifications to the paper:

* We have added experiments with varying missing rates (Appendix E)
* We have added a new appendix (Appendix D) that provides more details about the theoretical properties of the variational bound (in particular, the regularity conditions for convergence of the bound).
* We have made a few small clarifications (see details in the individual responses)

---

### Decision · Program_Chairs · 2021-01-07
**Final Decision**

**Decision:**

Accept (Poster)

**Comment:**

All the reviewers highlight that the paper addresses the important issue of extending deep latent variable models to handle missing non at random data, which are known to be very difficult. The authors suggest modeling the mechanism of missing values and perform inference  using amortized importance weighted variational inference and demonstrate the capacities of their approach on many experiments. The paper highlight the trade-off between the complexity of the data model and that of the missing data mechanism. The authors appropriately answer reviewers comments, add new experiments varying the percentage of missing values, and give more details on the methodological part.
I also think that this is a valuable contribution to the community, that the literature is well covered (the historical statistical litterature and the ML one), and that it provides new insights and methods to tackle this difficult problem.